# Lineage tracing using a Cas9-deaminase barcoding system targeting endogenous L1 elements

Byungjin Hwang[1], Wookjae Lee [1], Soo-Young Yum [2], Yujin Jeon [1], Namjin Cho[1], Goo Jang [2] &
Duhee Bang[1]

Determining cell lineage and function is critical to understanding human physiology and pathology. Although advances in lineage tracing methods provide new insight into cell fate, defining cellular diversity at the mammalian level remains a challenge. Here, we develop a genome editing strategy using a cytidine deaminase fused with nickase Cas9 (nCas9) to specifically target endogenous interspersed repeat regions in mammalian cells. The resulting mutation patterns serve as a genetic barcode, which is induced by targeted mutagenesis with single-guide RNA (sgRNA), leveraging substitution events, and subsequent read out by a single primer pair. By analyzing interspersed mutation signatures, we show the accurate reconstruction of cell lineage using both bulk cell and single-cell data. We envision that our genetic barcode system will enable fine-resolution mapping of organismal development in healthy and diseased mammalian states.

[1] Department of Chemistry, Yonsei University, Seoul 03722, Korea. [2] Laboratory of Theriogenology and Biotechnology, Department of Veterinary Clinical Sciences, College of Veterinary Medicine, the Research Institute of Veterinary Science, and BK21 PLUS Program for Creative Veterinary Science Research, Seoul National University, Seoul 08826, Korea. These authors contributed equally: Byungjin Hwang, Wookjae Lee, Soo-Young Yum. Correspondence and requests for materials should be addressed to G.J. (email: snujang@snu.ac.kr) or to D.B. (email: duheebang@yonsei.ac.kr)

Understanding the history of a cell is attractive to developmental biologists and genetic technologists because the lineage relationship illuminates the mechanisms underlying both normal development and certain disease pathologies. Researchers have developed a vast arsenal of robust genomic tools to interrogate cells. Traditionally, determining the history of individual cells has been accomplished using fluorescent proteins[1], Cre-loxP recombinase[2], somatic transposon events[3], and accumulated microsatellite mutations[4]. More recently, various cellular barcoding strategies have been developed[5–9]. To this end, the clustered regularly interspaced short palindromic repeats (CRISPR)/Cas9 genome editing system[10–13] has been primarily used to develop many cellular barcoding methods. However, creating repeat copies of array elements is difficult using methods that incorporate the CRISPR/Cas9 system, resulting in the need to introduce multiple exogenous DNA barcoding array elements into the genome to create a stable transgenic line for model organisms.

The majority of the human genome consists of repetitive elements that have been utilized for live imaging[14] and fetal aneuploidy testing[15]. Recently, researchers have shown that the editing of multiple endogenous retrovirus genes can be achieved without altering normal development[16]. Inspired by this evidence, we reasoned that multiple target regions in various interspersed sites in repeat elements could serve as barcodes for lineage analysis. We hypothesized that CRISPR/Cas9-induced double-strand breaks (DSBs) in these numerous endogenous interspersed sites present in the genome would result in the loss of useful information. Thus, we used the recently proposed base editing method involving nickase Cas9 (nCas9) fused with cytidine deaminase (referred to as BE3 in previous literature[17] but hereafter referred to as targeted deaminase), which converts C:G base pairs to T:A

base pairs in the 4–8 nucleotide region from the protospacer adjacent motif (PAM) on the distal side of the protospacer sequence without inducing DSBs.

Consequently, we developed a new cellular barcoding method for lineage tracing using nCas9 fused with cytidine deaminase to target the long interspersed nuclear element-1 (L1) in the genome. We believed that the interspersed target regions of the targeted deaminase system could be leveraged to introduce various substitution patterns into the regions using only a single-guide RNA (sgRNA). We confirmed that these unique marker combinations were gradually introduced and that tracking the patterns enabled the accurate reconstruction of the lineage relationship. In addition, we validated our approach at the single-cell level using time-lapse imaging experiments as ground truth data. We expect that our genetic barcoding system could provide insight into normal cell development as well as molecular pathology.

## Results

**Characterization of target regions for cell barcoding system.** First, we evaluated the amplified regions that were known to be human L1 retrotransposon regions[15]. A single primer pair was used to amplify the L1 retrotransposon region (Fig. 1a and Supplementary Figure 1a). The region was designed to maximize the number of distinct sequences and to increase the potential for uniform amplification[15]. We reduced PCR amplification bias by using two-step PCR with a primer containing degenerate bases according to a principle similar to that of a previous method (see Methods)[18]. The amplicon size (measured in fragment length) was bimodally distributed, with 99% of the regions overlapping with known L1 subfamilies (Supplementary Figure 2,

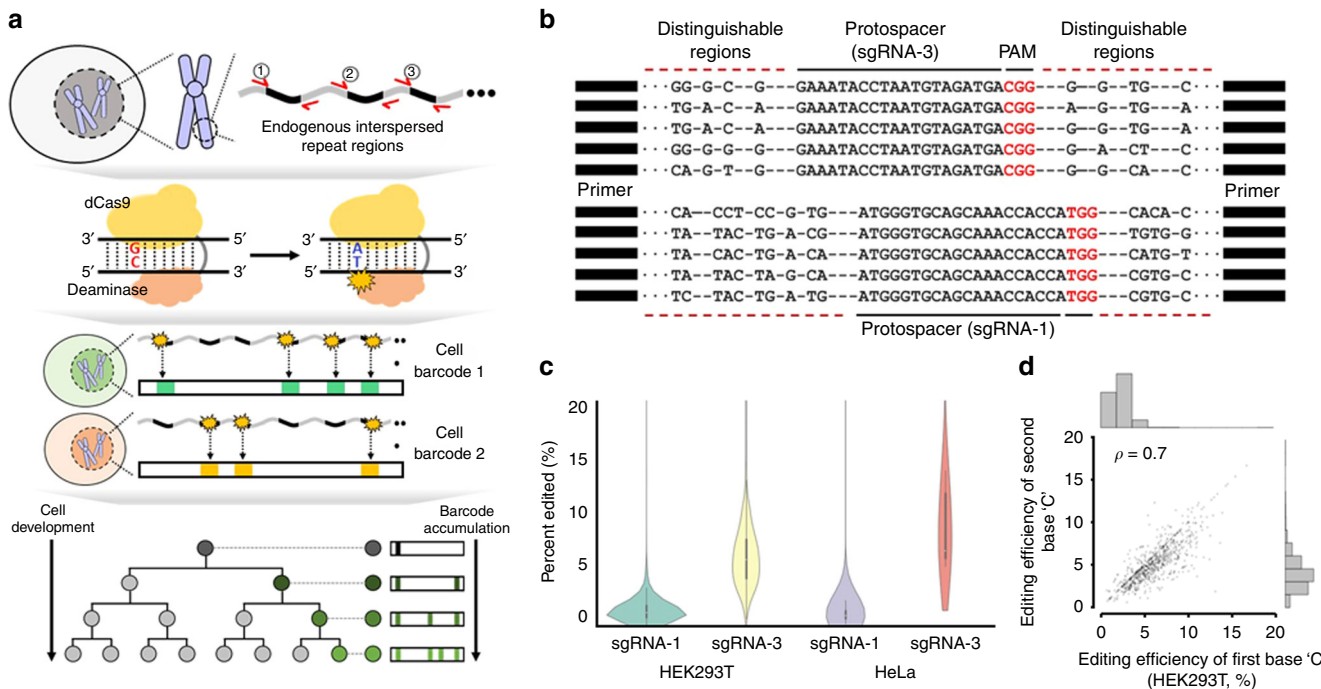

**Fig. 1** Targeted deaminase system for lineage tracing. **a** Schematic overview of the targeted deaminase system. Red arrows indicate the single flanking primer pair used to amplify the targeted regions. The substitution pattern in the target region served as the cell barcode for tree reconstruction. **b** Pairwise alignment of five representative target sites for sgRNA-1 and sgRNA-3 design. These regions were amplified using a single primer pair and alignment distinguished each target site by the different surrounding sequences. **c** Editing efficiency of the two selected sgRNAs in the target region for HEK293T and HeLa cells. **d** Correlation of editing efficiency between the first and second base C in a specific window (4–8) within the sgRNA-3 spacer sequence in HEK293T cells (Spearman's correlation = 0.6 for HeLa cells). The source data are available in the Source Data file

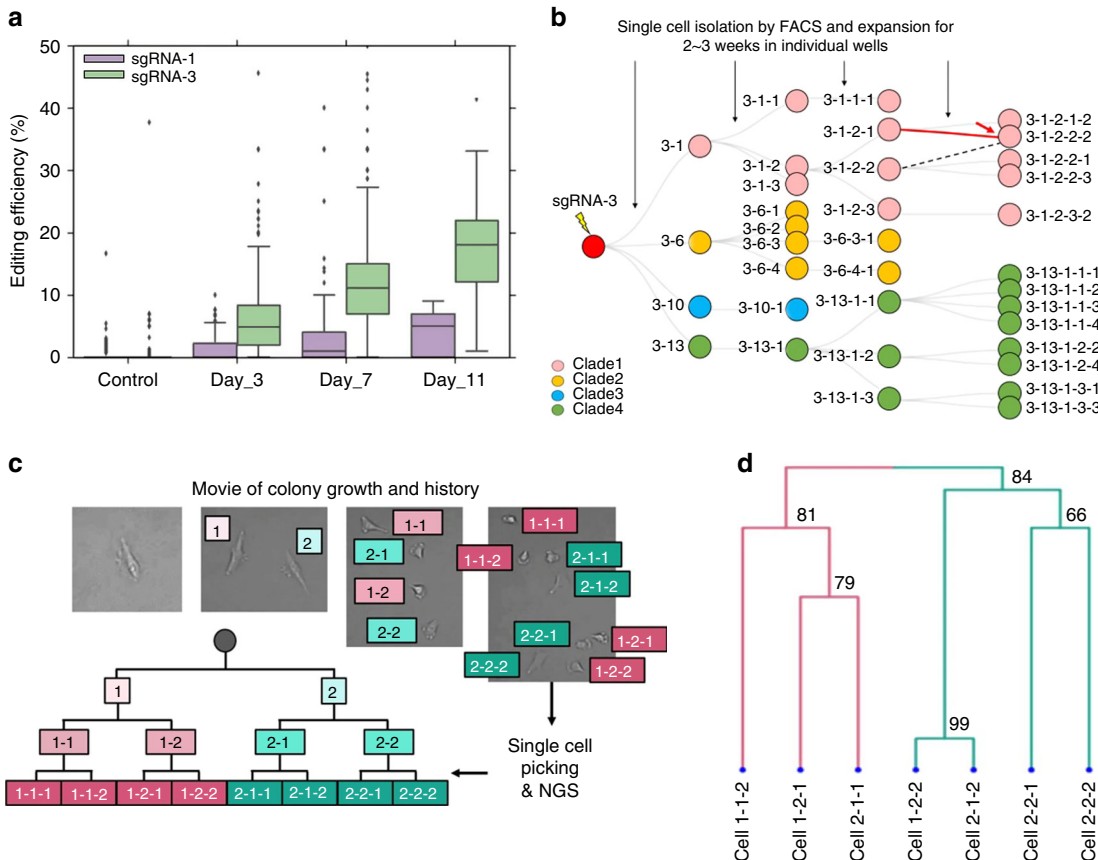

**Fig. 2** Cumulative editing and analysis of lineage tracing experiment. **a** Cumulative editing efficiency in the target regions using sgRNA-1 and sgRNA-3 for the defined time points. **b** A representation of the tree experiment for sgRNA-3 in HEK293T cells. The single-cells were sorted and expanded to a limited number of cell divisions. The bulk cells were then amplified from a 96-well plate and sequenced for cell lineage analysis. A network graph[7] was constructed from all the pairwise cell barcode information. The "−" represents the mother-child node relationship and the colors of each circle represent the different clade (pink: Clade1, yellow: Clade2, blue: Clade3, green: Clade4). The misplaced nodes are represented by a red arrow. The red solid line connects the incorrectly placed mother-daughter node, and the dotted line indicates the correct mother-daughter node connection. **c** Time-lapse imaging of the single-cell expansion experiment. At the end of the video, single-cells were picked and sequenced for lineage reconstruction. **d** An example of a tree expansion experiment for seven single-cells. The bootstrapped confidence score (bootstrapped P-values × 100, n = 1000) are shown at the branching point. Physical isolation was impossible for the left most cell (1–1–1) in (**c**). The source data are available in the Source Data file

see Methods). To compare the effect of precise molecular tag counting, we compared the effect of introducing degenerate bases (unique molecular identifier, UMI) at the 5′ end of one of the primers. An average of 30% of tags (with at least 50% of the terminal base sequences of the amplification primer being correct) were uniquely aligned to a reference genome, and we obtained 25,933 and 26,347 distinct aligned positions (average number of uniquely aligned positions from four replicate experiments for non-barcode and barcoded samples, respectively). The uniformity (coefficient of variation) was higher in the barcoded sample (0.58 compared to the 0.82 for non-barcoded), minimizing the possibility of over-counting the true number of molecules in the final tally for each target region.

Next, we searched for the endogenous regions that could serve as barcodes after genome editing using targeted deaminase. Like the existing CRISPR/Cas9 system, targeted deaminase targets a 20 bp spacer adjacent to the NGG PAM sequence and induces C-to-T conversion in the specific protospacer sequence window. For proof-of-concept, we screened all targetable spacer regions to identify those with the NGG PAM sequence in the retro-transposon region and compiled the candidate spacer regions with a C in the 4–8 nucleotide window from PAM on the distal side of the protospacer. We confirmed that there were several

sgRNA candidates with identical spacer sequences that satisfied our established conditions. There were 126,810 targetable L1 elements in our design (PCR with a pair of primers). Of these, 17,956 spacers had at least two perfect matched sites in the targetable L1 elements. In addition, when we investigated the top 100 targetable L1 elements, the average pairwise similarity was 28% and each had a high number of perfectly matched sites (range: 95–1585). We then selected two spacer sequences with the highest number of precisely matched sites (top two, highest ranking [sgRNA-1] and third-ranked [sgRNA-3]) in the targeted repeat regions (Supplementary Figures 1b and 3). The sequences of sgRNA-2 and sgRNA-1 were nearly identical (only a single base difference); thus, sgRNA-2 was omitted from further experiments. We expected that one sgRNA would introduce a substitution at multiple target sites in the L1 regions and defined cell barcode as C-to-T substitutions (see Methods). Although the multiple target sites of each sgRNA had identical spacer sequences, they were distinguishable after amplification of the target region via the surrounding sequences, which could be uniquely aligned to the specific genomic position (Fig. 1b).

**Analysis of genome editing in repeat regions.** We first applied the targeted deaminase system (Supplementary Note and

Supplementary Figure 4) on L1 retrotransposon regions in HEK293T and HeLa cells to test whether the system could be used in lineage tracing experiments. When we analyzed multiple target sites (targeted by a sgRNA-1) with identical spacer regions, the average editing efficiencies (fraction of the number of C>T reads) for HEK293T and HeLa cells were 1.5% and 2.3%, respectively (114 and 143 cell barcodes, respectively), for multiple targets in the known 4–8 nucleotide spacer sequence windows (Fig. 1c), implying a low number of edited cells. In contrast, sgRNA-3 exhibited an average 4-fold higher editing efficiency (6.3% and 9.3% in HEK293T and HeLa cells, respectively) compared with sgRNA-1, and the correlation of the editing efficiency between the two Cs in the spacer sequence windows were high in the multiple targets (Fig. 1d) which means base editing that converts most Cs in the editing window is processive. Thus, we treated these two Cs as a single editing site, by averaging the editing efficiency for the lineage tracing. We also observed differences in the editing efficiency between the two cell lines across the target sites. No significant background mutations were detected compared with the vehicle control without targeted deaminase and sgRNA (P-value < 2.2e −16, Mann–Whitney U-test). As well, aside from Cs in the known 4–8 nucleotide window in the PAM distal end, very low editing frequency of non-target Cs was observed (P-value < 2.2e −16, Mann–Whitney U-test, Supplementary Figure 5).

**Cumulative introducing of cell barcodes by targeted deaminase**. Next, we investigated whether the targeted deaminase system could continuously introduce genetic barcodes in the target repeat regions. To explore the extent of the continuous editing strategy at known time points, we compared sgRNA-1 and sgRNA-3 via repeat serial transfection of the targeted deaminase system at approximately 3-day intervals (Fig. 2a). The average editing efficiency increased linearly with the gradual accumulation of edited sites following serial transfection, and the observed editing rate was faster using sgRNA-3 compared with sgRNA-1. Thus, we concluded that the continuous introduction of genetic cell barcodes was feasible using our method.

**Lineage tracing using a controlled in vitro bulk experiment**. We devised an in vitro cell expansion experiment to investigate whether the cell barcodes were properly introduced each generation for the tree construction. HEK293T and HeLa cells were transfected using the PiggyBac™ transposon system containing the mCherry fluorescence protein, targeted deaminase, and sgRNA (Supplementary Figure 4). After sorting by FACS into 96-well plates, single-cells from the two cell lines underwent clonal expansion. The mCherry-positive single-cells were then sorted and cultured in different wells, and the process was repeated (Supplementary Figure 6). The target regions from the expanded cells were amplified using a single primer pair and subjected to next-generation sequencing (NGS). The system was first applied to the HEK293T cells. The known tree topology within each node (a node represents the sum of the cell barcodes) represented by the bulk cells enabled the validation of the lineage tracing. On average, 95% of the unique reads were aligned and these reads were processed further as multiple alignments can occur because of homology between the flanking regions. After alignment and variant calling, we detected an average of five cell barcodes per node in the first generation of the tree and found that ~93% of the target sites were unedited by sgRNA-1.

Next, tree building was performed using an iterative graph approach[7] and an additional post-hoc cell barcode selection step (see Methods). For sgRNA-1, the tree could not be

identified correctly because of a low number of informative cell barcodes. Conversely, sgRNA-3 showed an average of 29 cell barcodes for the first generation nodes. The accuracy of the reconstructed tree was defined based on the fraction of the correct node placements on the known depth and position of the tree. For sgRNA-3, we achieved 81% accuracy (29/36) of the reconstructed tree based on the variant calling approach. To remove spurious connections and refine the reconstructed tree, we developed a custom algorithm to obtain confident cell barcodes. We used a probabilistic approach with a posterior calculation to select the final candidate cell barcode for the tree construction (see Methods). Compared to the conventional variant calling approach, we observed an average of 70 cell barcodes for sgRNA-3 and the reconstruction accuracy improved to 97% (35/36) (Fig. 2b). For the HeLa cells, we also observed slightly improved performance in accuracy (88 and 97% (59/61) for conventional graph searching and variant calling vs. the custom algorithm, respectively) (Supplementary Figure 7). We note that the remaining errors might confound lineage reconstruction due in part to the nature of the bulk sequencing experiment, because some PCR or sequencing errors could have contributed to the final barcode combination of a given specific node (see Supplementary Figure 8 and Methods).

**Lineage relationship reconstruction at the single-cell level**. Next, we explored whether our targeted deaminase system could be extended to the reconstruction of lineage relationships at the single-cell level. Because the tree reconstruction efficiency was better with sgRNA-3 than sgRNA-1, we focused on sgRNA-3 only and chose HeLa cells for the single-cell level lineage tracing for the ease of isolating single-cells. HeLa cells were transfected with the PiggyBac™ transposon system containing the mCherry fluorescence protein, targeted deaminase, and sgRNA-3. We used time-lapse imaging to generate ground truth tree data and picked individual cells (n = 32 total single-cells analyzed [3 different trees]) determined to be mCherry marker-positive by manual inspection (Fig. 2c). To prepare for the sequencing experiment, we first performed whole-genome amplification (WGA) and subsequent PCR amplification of the selected single-cells. However, uneven distribution of the sequencing reads prohibited robust cell barcode identification. Furthermore, it has been reported that increased background C-to-T mutations can occur because of the high denaturation temperature during WGA[19]. Thus, we elected to optimize the single-cell PCR conditions. After the optimization, we achieved more uniform distribution of depth over the target regions (Supplementary Figure 9 and Methods). On average, 92.1% [0.3 standard deviation (stdev)] was covered for all single cells amplified by PCR. Due to the stochastic nature of the edits, the number of edited sites varied from 6.4 to 20 (0.8–2.4%, stdev; range 3.3–9.3). A standard agglomerative hierarchical clustering approach was used for three different experimental trees for three to four divisions (8–16 cells) of the expansion. Confident cell barcodes were encoded in the binary state and pairwise cell-to-cell distances were calculated for the tree reconstruction. Hierarchical clustering using an average-linkage method consistently recovered the known trees validated in the imaging experiment (Fig. 2d and Supplementary Figure 10) with a high computed Cophenetic correlation (0.92, 0.91, and 0.81).

To determine if the editing rate could affect the accuracy of the lineage reconstruction generated using our platform, we first estimated the accumulated mutation rate by approximating the editing dynamics of sgRNA-3 (Supplementary Figure 11). After exponential fitting, a rate of 0.06 edits per hour accurately reflected the experimental editing dynamics. Using this parameter

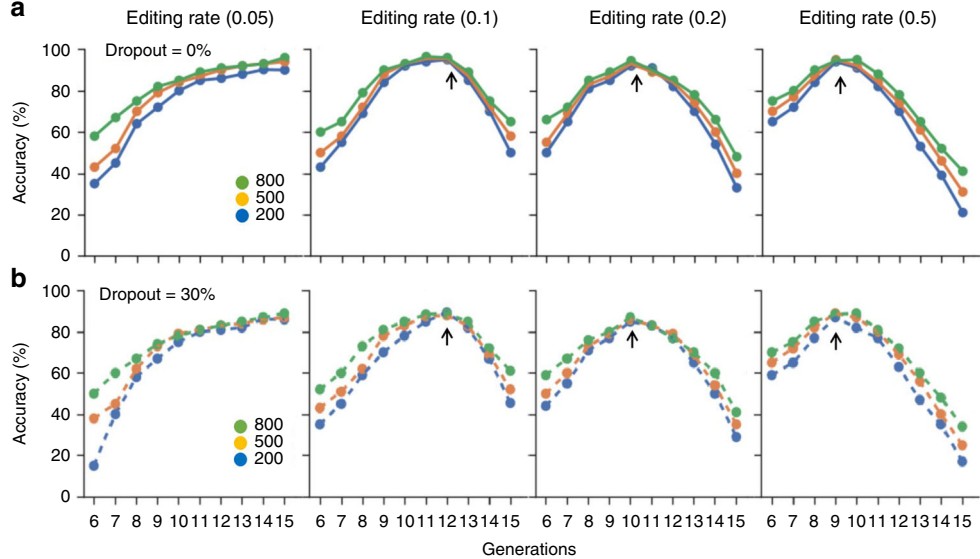

**Fig. 3** Performance simulations of the targeted deaminase system. Continued editing was allowed until G generations (see Methods for detail). After that, the accuracy (y-axis) was calculated as the percentage of correct order of pairwise relationships between the edits generated within G generations (see Methods). We compared the effect of dropouts on accuracy. **a** No dropout shown by solid lines and (**b**) dropout applied shown by dashed lines. The inflection point is indicated by the black arrow. Different colors correspond to different numbers of target sites and the units for the editing rate (base substitution rate) are mutations/site/cell division. The source data are available in the Source Data file

as the rate of editing efficiency, we conducted a simulation to predict the number of cell barcodes per cell expansion using an empirical number of target sites based on the experimental results. The simulation results indicated that the number of barcodes generated at each depth of the tree accurately reflected the experimental results obtained from the single-cells (comparison of the average number of cell barcodes per cell; experimental vs simulation; Tree 1: 6.4 vs 6.1, Tree 2: 20.3 vs 18, Tree 3: 9 vs 9.5, Fig. 2d and Supplementary Figure 10). Using a model that included the number of target sites and editing rate as variables, we found that the accuracy (Cophenetic correlation) improved with an increasing number of target sites and editing rate (Supplementary Figure 12).

**Lineage tracing simulation using targeted deaminase system.** We conducted simulation experiments to check the validity of our method for long-term lineage tracing (Fig. 3, see Methods). First, we performed simulation experiments assuming that dropouts occurred as the editing continued until a given generation (x-axis). The reconstruction accuracy increased as the number of traceable generations increased for a given number of editing sites. For the earlier generations (up to the 9th generation) before it reaches the peak, we observed a sharp increase in accuracy because the process of accumulating barcodes was sparse, which means that there was insufficient sharing between sister clades (two descendants that split from the same node) to allow accurate placement. Moreover, as the process of mutation accumulation is exponential, when the fraction of shared barcodes between sister clades exceeded ~85%, the increase in accuracy was minimal (<2%) before reaching the peak. It reaches a peak when the editable sites were almost used up. At this turning point, >70% of the editable sites were consumed for all analyzed cells, resulting in ambiguous conclusions about phylogenetic relationships after this peak point and diminishing returns. Notably, if the editing rate increased, this peak shifted to an earlier generation. In contrast, slower editing rate was not saturated until the last generation. Also, increasing the number of

targetable loci improved the reconstruction accuracy. After applying a 30% dropout rate across the target, the accuracy dropped by an average of 6.5% ($F = 0.05$ (editing rate), close to our experimentally determined value of 0.06), 6.7% ($F = 0.1$), 6.6% ($F = 0.2$), and 5.7% ($F = 0.5$), respectively, for each editing rate. When we further simulated the effect of site editing bias (Supplementary Figure 13), we observed no difference or a decreased difference (<1%) in accuracy.

**Discussion**

Here, we demonstrated a lineage analysis platform using targeted deaminase with sgRNA. Further, we showed that library preparation could be simplified using a single flanking primer pair to amplify the edited endogenous repeat element (L1 retrotransposon region[20]). As well, our method relies on more predictable barcodes based on substitution patterns. Most importantly, our method does not rely on introducing exogenous barcodes without making dsDNA breaks. We used endogenous repeat elements as potential barcodes, eliminating the need to create and insert complex repeat arrays. The genetic editing pattern in endogenous sites can serve as a cell barcode to reconstruct the cell lineage tree. A limitation of conventional Cas9-induced lineage tracing is the rapid saturation of genetic barcodes, which might hinder long-term labeling experiments. From the perspective of long-term lineage tracing (0.06 edit per hour for our method vs 0.4 edit per hour for the conventional Cas9-induced method), our approach enables the editing of targets slowly but continuously, because PiggyBac-based plasmid delivery lasts longer inside the cells. However further studies that provide a direct comparison to previously established barcoding systems using sgRNA-Cas9 ribonucleoprotein are needed. The simulation experiments provided the rationale for long-term lineage tracing and showed a robust performance even when site editing bias was present at the targets. When considering dropout effects in a simulation study, these findings illustrate that lineage tracing with our base editing system may have an advantage over conventional Cas9-based system.

Notably, the high theoretical diversity (sgRNA-3, $10^{250}$) of the cell barcodes is sufficient to cover approximately 37 trillion cells in the human body. Although we have only used targeted deaminase for the lineage analysis, the mutation rate could be controlled using other base editing systems[21,22] or the rational design of sgRNA that target other regions. In addition, other engineered dCas9 proteins or other classes of CRISPR proteins[23–26] might influence the editing efficiency or target diversity. Alternatively, we could simultaneously integrate multiple sgRNAs into the genome and track the resulting mutation patterns to reconstruct the lineage in a more predictable manner.

Our method has some technical challenges. Although the potential diversity of our proposed method is substantially larger than the diversity achieved using previous methods, we could not fully explore this aspect because of the limited number of single-cells sampled and the low editing rate. In the conventional Cas9 system, edited barcode arrays are amplified and can be readout by paired-end sequencing[5]. Conversely, clonal information could not be directly inferred from our bulk cell analysis because of the interspersed nature of the barcode regions. The analyzable clonal sequence space is likewise limited by current sequencing length (300 bp for the Illumina platform using the 150 bp paired-end mode). Instead, we devised a lineage relationship between the bulk cells (isolated from a single well) that could be determined based on the integrated cell barcode pattern of the cell population. We acknowledge that our current platform has been tested over a limited number of generations (up to four generations). In the in vitro tree expansion experiment, we observed some errors (~3%) in the reconstruction of the tree. However, we anticipate that this negative effect on reconstruction accuracy would be attenuated by simultaneously introducing multiple sgRNAs into the interspersed target sites, as this would increase the complexity of the barcodes and potentially facilitate lineage reconstruction for multicellular higher organisms such as mice. Furthermore, targeted deaminase can be cytotoxic if continuously expressed within cells, and this requires additional investigation.

We employed a custom algorithm to select robust cell barcodes for bulk cell-based lineage tracing because conventional variant calling based on the stringent FreeBayes algorithm returned a small number of variants. This allowed flexible control of the various parameters and eliminated the need for simple, sequencing depth-based filtering. In a previous study, the original tree topology was preserved with high accuracy using a graph-based approach[7]. In our single-cell analysis, tree building for three different sets of three to four generation single-cell division was also accurate. In simulations, we detected higher accuracy of the tree reconstruction with an increasing number of barcodes. In a practical scenario in which conventional Cas9 causes barcode dropout, our base editing scheme (without dsDNA breaks) could potentially be useful for lineage construction. In the future, additional comparative studies are required to investigate whether the base editing strategy confers an advantage over the Cas9 nuclease-based method from the perspective of the effect of barcode dropout on tree reconstruction.

In summary, we have shown a proof-of-concept for using the targeted deaminase system for lineage tracing through in vitro tree expansion experiments and time-lapse imaging. In doing so, we have laid the foundation for developing an alternative technique to trace cell lineages using endogenous targets. In future studies, we hope that our system can be used to determine the fates of different cells for organ development in a transgenic model organism. In the long-term, if coupled with transcriptional information, our method could provide a high-resolution view of developmental lineages.

## Methods

**Plasmid construction**. The PB CMV-BE3 EF1α-mCherry-T2A-puro sgRNA (Supplementary Note) was constructed by PCR assembly of the U6-sgRNA expression cassette from the gRNA_Cloning Vector (Addgene plasmid #41824, Addgene, USA), whereas CMV-BE3 was constructed from pCMV-BE3 (Addgene plasmid #73021), the puromycin gene from lentiGuide-Puro (Addgene plasmid #52963), and mCherry from PB_tet_attB-mCherry (a gift from Seung Hyeok Seok in Seoul National University College of Medicine) on a PiggyBac transposon backbone from PB-CA (Addgene plasmid #20960). The targeting sgRNA sequences were cloned according to the provided hCRISPR gRNA synthesis protocol. The pCy43 PiggyBac Transposase vector was provided by the Sanger Institute (Hinxton, UK). All cloned plasmids were confirmed by Sanger sequencing. Plasmids were prepared using the EndoFree Plasmid Maxi Kit (QIAGEN, USA) and Exprep Plasmid SV kit (GeneAll, Korea) according to manufacturers' protocols.

**Cell lines and cell culture**. All cell lines were obtained from the KCLB (Korean Cell Line Bank) and maintained at 37 °C with 5% $CO_2$. The HEK293T human embryonic kidney and the HeLa human cervical cancer cell lines were cultured in Dulbecco's Modified Eagle's Medium (DMEM; Gibco, USA) supplemented with 10% fetal bovine serum (FBS; Gibco, USA) and 1% penicillin/streptomycin (P/S; Thermo Fisher Scientific, USA). All cell lines were tested for mycoplasma and confirmed that there was no mycoplasma contamination in the cells.

**Generation of cells containing the genetic barcoding system**. To generate cells harboring genetic barcodes, HEK293T and HeLa cells were individually transfected with genetic barcoding system plasmids at a 2:1 transposon (PB CMV-BE3 EF1α-mCherry-T2A-puro sgRNA) to transposase ratio using Lipofectamine™ 3000 (Life Technologies, USA) according to the manufacturers' protocols. The transfected cells were incubated for approximately 3 days and harvested for genomic DNA (gDNA) using the DNeasy Blood & Tissue Kit (QIAGEN, USA).

The experiment that entailed the continuous introduction of genetic barcodes was conducted using the aforementioned transfection method. Half of the cells were harvested for gDNA 3 days after transfection and the other half of the cells were cultured for the serial transfection.

In the experiment to measure barcode editing efficiency, only the genetic barcoding system without transposase was transfected using Lipofectamine™ 3000. The transfected cells were harvested at 4, 8, 12, 16, 20, 24, 30, 36, 42, and 48 h, followed by gDNA extraction.

**A controlled in vitro tree experiment**. The procedures used to generate the barcodes were identical to the protocols described above, and successfully transfected cells were selected using puromycin (2 µg/ml). The selected cells were isolated by single-cell FACS using an Aria II FACS machine based on mCherry fluorescence (a marker of transfection). The sorted single-cells were then cultured in DMEM supplemented with 20% FBS and 1% P/S. Single-cell-derived populations were expanded in culture for up to 3 weeks. Half of the expanded cells were harvested and gDNA was extracted for NGS as described in the Methods (Endogenous genetic barcode amplification—bulk cells) section and the remainder of the cells from each lineage were pelleted and frozen at −20 °C.

**Endogenous genetic barcode amplification in bulk cells**. The gDNA extracted from the cells described above were used for the amplification of endogenous genetic barcodes. Kapa High Fidelity Polymerase (Kapa BioSystems, USA) was used for all barcode amplifications. Up to 500 ng of gDNA was loaded into a single 20 µl initial PCR reaction that included 1 µl of 10 µM forward and reverse primers (L1 site for and L1 site rev in Supplementary Table 1), 10 µl KAPA DNA polymerase, and up to 8 µl of nuclease-free water and amplified using primers with a sequencing adaptor according to following protocol: 98 °C for 120 s followed by two cycles of 98 °C for 10 s, 57 °C for 120 s, and 72 °C for 120 s and a final 10 min step at 72 °C. Homemade AMPure XP beads[27] (hereafter, AMPure beads) generated using Sera-Mag SpeedBeads (6515-2105-050350, Thermo Scientific, USA) were then used to purify the initial PCR products. The initial PCR products were then loaded into a single 20 µl second index PCR reaction and amplified with index primers using the following protocol: 30 s at 98 °C followed by 15 cycles of 10 s at 98 °C, 30 s at 60 °C, and 30 s at 72 °C and a final 10 min step at 72 °C. The second PCR product was then purified using 1.2× AMPure beads. All primers were prepared by IDT (Integrated DNA Technologies, USA). The sequencing was performed on the Illumina NextSeq 500 system using a NextSeq 500/550 High Output v2 kit (300 cycles) (Illumina, USA). The sequencing statistics for the bulk experiments are provided in Supplementary Table 2.

**Transfection for time-lapse imaging**. The HeLa cells were prepared by subculturing in a 35-mm dish with 2 mL DMEM, supplemented with 1% P/S, 1% non-essential amino acids (NEAA) (Gibco, USA), 100 mM 2-mercaptoethanol (2-ME) (Sigma-Aldrich), and 10% FBS. The cells were transfected with plasmids using an electroporation system (Neon®, Invitrogen, voltage:1140v, pulse width

range: 40 ms, pulse number: 1). After 4 h of transfection, the culture medium was replaced with fresh medium to remove the dead cells.

**Manual cell picking and monitoring for time-lapse imaging**. All of the single-cell manipulations were conducted using a micromanipulator device (Nikon-Narishige, Tokyo, Japan) during observation under an inverted microscope. One day after transfection, the cells were trypsinized and washed in phosphate-buffered saline (PBS; Gibco, USA). To manually pick single-cells, the cell suspensions were placed in a drop of PBS with 0.5% FBS and covered by mineral oil (Sigma, USA). A single RFP-positive only cell was aspirated using a micro-injection pipette (diameter: 20 μm, ORIGIO, Charlottesville, VA) under fluorescence exposure. The aspirated single-cell was transferred to a 4 μl droplet of DMEM supplemented with 1% P/S, 1% NEAA, 100 mM 2-ME, and 10% FBS in a 100-mm dish and overlaid with mineral oil at a ratio of one cell to each droplet. The single-cells were cultured for 4 h in a $CO_2$ incubator at 37 °C and then moved to an incubator equipped with a JuLI™ Stage real-time cell history recorder (NanoEnTek), which enables the observation of cell growth using live, time-lapse imaging.

**Endogenous genetic barcode amplification in single-cells**. Whole-genome multiple displacement amplification (MDA) of the single-cells (from manual cell picking) was conducted using the illustra™ Ready-To-Go GenomiPhi V3 DNA Amplification Kit (GE Healthcare, USA). MDA was performed according to manufacturers' protocols with the slight modification of increasing the reaction time from 1 h 30 min to 3 h. Next, 5 μl of the single-cell MDA product was added to a 20 μl initial adaptor PCR reaction that included 1 μl of 10 μM forward and reverse primers (L1 site for and L1 site rev in Supplementary Table 2), 10 μl KAPA DNA polymerase, and up to 8 μl of nuclease-free water. The PCR reaction was performed using the following protocol: 2 min at 98 °C followed by 10 cycles of 10 s at 98 °C, 2 min at 57 °C, and 2 min at 72 °C and a final 2 min step at 72 °C. After purification using AMPure beads, the initial PCR product was loaded into a second index PCR reaction and PCR was performed using the following protocol: 30 s at 98 °C followed by 15 cycles of 10 s at 98 °C, 30 s at 60 °C, and 30 s at 72 °C and a final 10 min step at 72 °C. The final product was then purified using AMPure beads.

Single-cell PCR amplification of the manually picked single-cells was performed using the PCR reaction composition ratio described above and following protocols: 1. Adaptor (initial) PCR: 2 min at 98 °C followed by 30 cycles of 10 s at 98 °C, 2 min at 57 °C, and 2 min at 72 °C and a final 2 min step at 72 °C. The product was then purified using AMPure beads. 2. Index (second) PCR: 30 s at 98 °C followed by 15 cycles of 10 s at 98 °C, 30 s at 60 °C, and 30 s at 72 °C and a final 10 min step at 72 °C. The products of the second PCR were then loaded onto a 2% agarose gel and separated by gel electrophoresis. The band at the expected size was purified using the QIAquick Gel Extraction Kit (QIAGEN, USA) according to the manufacturers' protocols. Likewise, sequencing was performed on the Illumina NextSeq 500 using the NextSeq 500/550 High Output v2 kit (300 cycles).

**Analysis of amplified post-alignment processing**. Reads were aligned to hg19 using BWA (v0.7.12-r1039) and realignment around indels (RealignerTargetCreator, IndelRealigner) was performed using GATK (v3.3-0). Per-position base calling was accomplished using the SAMtools (v1.1) *mpilup* function and the pileup file was used for custom variant calling (details in the next section). The aligned regions were annotated using RepeatMasker (http://www.repeatmasker.org) and the sizes of the amplified regions were plotted to calculate the overlap fraction.

**Accurate molecule counting to reduce PCR amplification bias**. For precise molecule counting, sequencing reads sharing the same UMI (degenerate bases) were grouped into families and merged if ≥70% contained the same sequence. In addition, to minimize the effect of over-counting the same molecules, we calculated the distances between UMIs; Hamming distances ≤2 were merged in the Hamming-distance graphs. We only retained UMIs exhibiting the highest counts within the clusters.

**Identification of confident sites for lineage reconstruction**. We first adopted a variant calling approach using FreeBayes (v1.1.0-3-g961e5f3) to extract confident markers (C>T substitutions) for the lineage reconstruction. The variant calling used FreeBayes (input from BAM after indel realignment) and filtered positions (depth >10) considered candidate markers, and only included the markers with higher allele frequency than the value calculated for the background control using an empty vector. For the bulk and single-cell linage tracing experiments involving HeLa cells, variant calling was performed using modified parameters (–ploidy 3, –pooled-discrete). To handle both the bulk and single-cell data efficiently, we developed a custom algorithm for a variant calling strategy that was based on our targeted deaminase system. We adopted a probabilistic approach using a binomial mixture model with conditional probabilities, as described in a previous study[28]. An expectation-maximization algorithm was used to estimate the model parameters to account for the inherent deviation of allele frequencies in unstable

genomes (e.g., genomes with different ploidies). Every candidate position in the target region, depth >10×, variant allele count >2, and posterior probabilities ≥0.95 was selected as a final marker. After performing a union operation for all the markers present in the bulk nodes, we selected confident markers using following criteria: First, we tabulated the distribution of the editing efficiencies of bulk cell lines across the target regions. Then, normalized the per edit site average editing efficiency to value of 1 by aggregating all sites and calculated the contributing fractions of each edited sites. These site edit probabilities (per site) were strongly correlated ($R^2 > 0.7$) between two cell lines. Since single-cells have different read structure compared to bulk cells, we designate the probability of bulk cells to single cells for the shared edit sites only. To reduce the false reconstruction of the lineage, we excluded the sites with an average editing efficiency greater than 80% (normalized site editing probability >0.004, ~10% of the target sites were removed) because they were mostly used up, in other words, shared by almost every clone, in lineage reconstruction. Also, we removed highly correlating features (or edited sites) ($R^2 > 0.8$) to reduce the chance of misplacing the nodes where two cells might accidentally share edits. The sum of these final markers for each cell represented the final cell barcode.

**Tree building for bulk and single-cell experiments**. For the bulk cell experiment, tree building was performed using an approach similar to the previous method (https://bitbucket.org/Bastiaanspanjaard/linnaeus). We used the base editing pattern to build the substitution graph. For simplicity, the nodes were identified as CIGAR string-like sequences (i.e., 1E10E means the first and the 10-th C position in the perfect on-target region was edited). The graph reconstruction strategy first employs a depth-first search (DFS) to identify the strongest connected components using edited sites. As opposed to the conventional algorithm used to identify connected components, we initially employed a DFS approach to maximize the weight of connected components (the sum of the sequencing depths of the component is maximized). As the graph search gives priority to high-depth components, the DFS-based algorithm resulted in a few nodes inadvertently being placed on different clades, as the depth of shared edits was unusually high. Therefore, we modified the algorithm to identify connected components based on the overlapping fraction of edited barcodes between nodes (nodes in the same clade share a higher fraction of barcodes than with nodes in other clades). In so doing, no errors were introduced in distinguishing the clades. Only minimal error (accuracy increased to 97% for experiments involving both HEK293T and HeLa cells) occurred in assigning the mother-daughter relationship within clades. Assigning the correct in-clade mother-daughter relationship depends largely on the continuous accumulation of edited barcodes. Thus, the remaining error appears to be due in part to the nature of the bulk sequencing results, in that some PCR or sequencing error contributes to the final barcode combination of specific nodes, thus leading to misassignment of the subtle mother-daughter relationship. For example, if a node (daughter) from one mother node got the edits that should belong to other node (daughter) from another mother node by chance resulting in more overlap with another mother node, the algorithm will likely place the node in the wrong place because assignment of the mother-daughter relationship depends on how much the barcode combination is shared with the ancestor.

Inherently, the first ancestor node (bulk cells) had a maximal degree of connections with the other nodes. We removed this node and identified the ancestor (top) node from the remaining connected components in an iterative fashion. To identify the top node for the remaining components, we must calculate the detection rate $p(x)$ first, assuming x is a top edit. This can be calculated as the ratio of the number of cells (nodes) that express edit x and an edit connected to $x$ to the number of cells (nodes) that express edits connected to $x \left[ p(x) = N_{x \cap C(x)} / N_{C(x)} \right.$, where $c(x)$ is a set of edits that are connected to edit x]. Thus, the chance of observing edits between edits x and y in at least one cell is calculated as $p(x - y) = 1 - (1 - p(x))^{N_y}$, where Ny represents cells (nodes) expressing edit y. As such, the underlying distribution of edit x is a *poisson binomial* with a different success probability defined as $p(x - y)$. Finally, we calculated the probability of measuring at most the observed degree using the distribution (if you have 4 nodes around the specific nodes but the actual observed degree is 3, then you calculate the cumulative distribution function for at most 3 connections). We used the *poissonbinom* R package to calculate the probability density. The node with the highest probability of this value is considered the top node (see Supplementary Figure 20a in ref. [7] (PMID: 29644996) for an illustrative example). This procedure was repeated until all the nodes were designated. Once all the pairwise cell networks were built, the cells were placed in the graph. We did not use the cell doublet detection threshold because scRNA-seq was not used in this study.

For the single-cell-based lineage tracing, the information was restricted regardless of whether the site was edited. To identify confident markers, blacklist candidate regions (integration of the single-cell results exhibiting no mCherry signal or vehicle control single-cells) were also filtered out. Unlike the bulk cell lineage construction, the time-lapse-based single-cell experiment contained the cells from the last depth of the expansion. Thus, the lineage tracing was accomplished using a different logic. The distance between the cells was calculated using the Jaccard index and hierarchical clustering was

performed using the *pvclust* function in R. Approximately unbiased probability values (P-values, bootstrap resampling) were calculated based on 1000 iterations.

**Inferring editing efficiency dynamics**. The HEK293T cells were transfected with the targeted deaminase vector (sgRNA-3). After 4, 8, 12, 16, 20, 24, 30, 36, 42, and 48 h post-transfection, the bulk cells were collected and amplified for sequencing (two replicates). Assuming a wild type fraction of 1 (100%) in $t = 0$, we estimated the editing rate by fitting the exponential function. We designated the wild type percentage as the fraction of unedited sites (C> T candidate positions) in the perfect on-target regions.

**Simulations**. We simulated the trees by varying different default parameters, such as tree depth (G), number of targeted sites (N), and the editing rate (F), assuming a stepwise constant rate for the accumulation of C>T conversion events across all targets. We chose the variables out of a wide range of variables for comparison with empirical estimates. We followed the tree depth until $G = 15$ [where we assumed a generation time of 1 day ($2^1 = 2$ cells)] because this is when our system seemed to reach saturation (Supplementary Figure 10). The editing rate is the probability of transforming the site to be edited ($F = 0.06$ for our study, the units for the editing rate (base substitution rate) are mutations/site/cell division). For all models, we allowed a model system to generate edits until G generations. Typically, within two weeks after fertilization (day 0), the cells of the epiblast begin to differentiate into three germ layers for human and mouse[29,30].

For a more realistic simulation and comparison to the previous approaches using conventional Cas9, we set up a scenario with a specific dropout rate (30%). We used binomial probability with N trials with a given $p = (1\text{-dropout}$ rate) to select the final editable sites. The dropout parameter reflects the effect of the nonhomologous end joining process, which could remove sequence between the cut sites if a conventional Cas9 system was used instead. Also, we applied site editing bias (10% of certain sites tended to saturate quickly). Biased editing simulates the differential editing efficiencies for different target sites [for simplicity, we rolled a dice from a uniform distribution and checked if each site was less than or equal to the $b = (1\text{-site editing bias})$]. This value ($b$) was determined to be the upper bound, based on the empirical distribution of site editing probabilities. Lastly, we simulated 800 targets (comparable to the sgRNA-3 design, 837 targetable sites) and applied the site editing bias. The accuracy of the simulation experiment was calculated as the fraction of triplets (a root node is divided into two daughters when the cell divides [the sum of mother and daughters]) that are correctly placed compared to the ground-truth mother-daughter relationship. For example, if we have a 3-depth tree ($1 + 2 + 4 = 7$ cells) with a total of 3 triplets and found 1 triplet that was correct, the accuracy would be 33% (1/3).

**Data analysis**. All statistical analyses were performed using the R computing environment and data plotting was conducted with the *seaborn* and *ggplot2* packages in R. For Figs. 1c and 2a, two-tailed Mann–Whitney U-test was used to test for significance of difference between the control and sample means.

**Reporting summary**. Further information on experimental design is available in the Nature Research Reporting Summary linked to this article.

## Code availability

All custom scripts used in making figures, downstream data processing and analyses are available at https://github.com/bjhwang113/BElineage.

## Data availability

The sequencing data supporting the findings of this study are available in the Sequence Read Archive with the identifier SRA SRP151792. A source data underlying Figs. 1c, d, 2a and 3a, b and Supplementary Figs. 2, 11 and 13 are provided as a Source Data file. A reporting summary for this Article is available as a Supplementary Information file. All other data are available from the authors upon reasonable request.

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

## Acknowledgements

This work was supported by: (i) the Mid-career Researcher Program (NRF-2018R1A2A1A05079172) from the National Research Foundation of Korea (NRF), funded by the Ministry of Science, ICT & Planning, (ii) the Bio & Medical Technology Development Program of the NRF, funded by the Korean government (MSIT; NRF-2016M3A9B6948494), and (iii) the Bio & Medical Technology Development Program of the NRF, funded by the Korean government (MSIT; NRF-2018M3A9H3024850).

## Author contributions

D.B., B.H. and W.L. conceived the project. B.H. analyzed the data with input from Y.J. W.L. performed the cell culture experiments with the help of N.C. S.Y. performed the time-lapse imaging experiment. B.H., W.L. and S.Y. wrote the manuscript with input from all coauthors. D.B. and G.J. supervised the project.

## Additional information

**Competing interests:** The authors declare no competing interests.

