## [Peer Review File · Nature Communications]

Reviewers' Comments:

Reviewer #1:

Remarks to the Author:

The authors describe a single cell lineage tracing system using BE3/targeted deaminase C>T base editing in L1 elements. In clonal expansions of HEK293T and HeLa cells they show their method can be used to reconstruct an accurate tree based on bulk and single cell experiments. I agree with their assessment of advantages (many target sites, slow editing) and disadvantages (sequencing of target sites cannot be incorporated in transcriptome-based sequencing methods, precluding large scale experiments). The paper is a useful characterization and test of a promising system in the field of single cell lineage tracing. However, the manuscript feels somewhat incomplete, and I would request additional analysis, as described in more detail below.

Major comments

- 1) The manuscript falls short on delivering strong proof that the system can replace current methods based on CRISPR/Cas9 lineage tracing with synthetic target sites. It would be important to generate and analyze deeper lineage trees (which would be more relevant for applications in multicellular organisms).
- 2) The authors should expand on the simulations (target sites + editing efficiency) in order to compare their system to current methods. Compared to CRISPR/Cas9 lineage tracing with synthetic target sites, the authors' system has more targets, but the information content of each target is very low (non-mutated/mutated). Using simulations, the authors should discuss how many targets n are needed to capture a theoretical tree of m cell divisions, and which experimental parameters are important (for instance, is there an optimal editing efficiency?).
- 3) How many L1 elements can the authors detect with their strategy, how many of those can be uniquely identified using the read sequence, and what fraction is typically detected in a single cell? It would be important to explicitly state these numbers to allow the reader to assess the advantages and limitations of the system. Please comment on the probability of the same site being edited independently in different cells.
- 4) Are there differences in editing efficiency between the different sites? This would be important to estimate the probability of the same site being edited in two different cells, which would complicate lineage tree reconstruction.
- 5) The authors should explain in the main text why they choose to use the iterative graph approach for the bulk experiments and hierarchical clustering for the single cell experiments.
- 6) The authors should indicate in figure 2b which three nodes are misplaced and explain why they think this happened.

Minor comments

- 7) The authors should explain which characteristics of L1 elements make this a good lineage tracing system, such as number of elements and sequence similarity between different elements. It's also unclear to me why the amplicon size was expected to follow a bimodal distribution (Results, first paragraph).
- 8) The figures should be improved to increase precision: Can you indicate cell sorting and approx. number of cell divisions in Fig. 2b? Why do the barcodes have different widths in Fig. 1a (orange and yellow bars)? And can you please highlight in Fig. 1a which sites are editable?
- 9) The authors show in Fig. 1d that the editing efficiency of the two editable Cs of sgRNA3 is highly

correlated, but they don't discuss the consequences or interpret this observation. Do you consider the two Cs in a target sites as separate entities for lineage tracing, or not?

10) How was the amplicon size measured?

11) Main text, line 87: Are these just mapped reads? How did authors control for PCR biases?

12) Main text, line 114: The experimental design is poorly explained, and it is not clear what process is repeated. How exactly does the sorting work? What is its purpose? Is it about monoclonal expansion of positive cells in order to reduce mosaic effects? If so, authors should clarify and justify the experimental design. Directing only to a tree (2b) is not informative.

13) Main text, line 181: The comparison of editing efficiency compared to Cas9 is not convincing. The authors compare to Cas9 injection into zebrafish embryos, which is a very different system, so it's likely that the difference in dynamics is not due to their system per-se but to the delivery of the recorder.

Reviewer #2:

Remarks to the Author:

The authors Hwang et al present here a straightforward application of the Cas9 deaminase system for use as a lineage tracing tool. The authors show in a stepwise fashion the progress from proof of concept in which they test by transient transfection two different gRNAs targeted to the L1 repeat element in the human genome, to application of this tool by PiggyBac transposons in bulk populations, and finally to single cells lineages. They use unique barcodes and a custom algorithm to build accurate trees, which they validate with "ground truth" time-lapse image data.

The authors' work is comprehensive, logically consistent, clearly described, and represents a step forward in the development of lineage-tracing tools. Comments below are minor in nature and could help the authors strengthen their work further.

1) How many L1 sites are available in the genome? Given the measured mutation rate, how many generations do the authors expect to be able to trace with this target set? It may also be useful to use their model to more generally show the number of generations traceable as a function of mutation rate and size of the target pool. Is there any risk, as the target space begins to saturate, that barcodes from different lineages will begin to collapse into the same, all-Cs-to-Ts in the same position at each target site, shared barcode?

2) Supplementary Figure 5 – please clarify by showing with a target sequence which Cs were or were not mutated.

3) The authors describe their system as dCas9-based in their abstract and introduction, but their vector map (Supplementary Figure 4) and the BE3 system originally described by reference #18 make use of nickase Cas9.

Point-by-point responses to reviewer comments:

Reviewers' comments:

Reviewer #1 (Remarks to the Author):

The authors describe a single cell lineage tracing system using BE3/targeted deaminase C>T base editing in L1 elements. In clonal expansions of HEK293T and HeLa cells they show their method can be used to reconstruct an accurate tree based on bulk and single cell experiments. I agree with their assessment of advantages (many target sites, slow editing) and disadvantages (sequencing of target sites cannot be incorporated in transcriptome-based sequencing methods, precluding large scale experiments). The paper is a useful characterization and test of a promising system in the field of single cell lineage tracing. However, the manuscript feels somewhat incomplete, and I would request additional analysis, as described in more detail below.

We appreciate the reviewer's important suggestion for improving our manuscript. We addressed the concerns that were raised and provided additional simulation results to strengthen the original manuscript with respect to comment #1.

Major comments

1) The manuscript falls short on delivering strong proof that the system can replace current methods based on CRISPR/Cas9 lineage tracing with synthetic target sites. It would be important to generate and analyze deeper lineage trees (which would be more relevant for applications in multicellular organisms).

We understand and take seriously the reviewer's concern about the validity of our proposed system that replaces the current CRISPR/Cas9 system with synthetic target sites. We agree that further experimentation to test our system in multicellular organisms would support the validity and advantages of our system. Because our manuscript is an initial communication of our findings, we believe that simulating deeper lineages with multiple parameters (e.g., number of target sites, dropouts, and differential detection probability) is sufficient to show that our system is an alternative method for lineage tracing studies. Our *in vitro* cell culture expansion experiment has analyzed deeper trees (four generations) in bulk experiments in comparison to published methods (PMID: 27918539, 27229144, three generations). In addition, the single-cell expansion experiment used three replicates for 3–4 generations (~8–16 cells). It was infeasible to perform further analysis and track single cells over four generations, due to the cells' strong tendency to form discrete clumps. When expanded cells exceed a certain number of generations, we are unable to manually pick or interrogate individual cells.

With respect, we believe that analyzing additional lineages will not benefit the testing of whether our system can be used for lineage tracing, over the existing cell line level methods. We hope that when we are able to apply our system to multicellular organisms, the term "deeper" from the reviewer #1 can be clarified. We are actively looking for funding to apply our system to the mouse model. Therefore, we believe that providing extensive simulation results can address reviewer #1's concern. Please refer to the response below (#2)

2) The authors should expand on the simulations (target sites + editing efficiency) in order to compare their system to current methods. Compared to CRISPR/Cas9 lineage tracing with

synthetic target sites, the authors' system has more targets, but the information content of each target is very low (non-mutated/mutated). Using simulations, the authors should discuss how many targets n are needed to capture a theoretical tree of m cell divisions, and which experimental parameters are important (for instance, is there an optimal editing efficiency?).

As per reviewer's suggestion, we have added extensive simulation results with Figure 3 and Supplementary Figure 12.

For a detailed description of the method, we added the sentences below on page 19, lines 435–457.

"We simulated the trees by varying different default parameters, such as tree depth (G), number of targeted sites (N), and the editing rate (F), assuming a stepwise constant rate for the accumulation of C>T conversion events across all targets. We chose the variables out of a wide range of variables for comparison with empirical estimates. We followed the tree depth until $G=15$ [where we assumed a generation time of 1 day ($2^1=2$ cells)] because this is when our system seemed to reach saturation (**Supplementary Figure 10**). The editing rate is the probability of transforming the site to be edited ($F=0.06$ for our study). For all models, we allowed a model system to generate edits until G generations. Typically, within two weeks after fertilization (day 0), the cells of the epiblast begin to differentiate into three germ layers for human and mouse (PMID: 29880675, 25349455).

For a more realistic simulation and comparison to the previous approaches using conventional Cas9, we set up a scenario with a specific dropout rate (30%). We used binomial probability with N trials with a given $p = (1-\text{dropout rate})$ to select the final editable sites. The dropout parameter reflects the effect of the nonhomologous end joining process, which could remove sequence between the cut sites if a conventional Cas9 system was used instead. Also, we applied site editing bias (10% of certain sites tended to saturate quickly). Biased editing simulates the differential editing efficiencies for different target sites [for simplicity, we rolled a dice from a uniform distribution and checked if each site was less than or equal to the $b = (1-\text{site editing bias})$]. This value (b) was determined to be the upper bound, based on the empirical distribution of site editing probabilities (see also related response below #3 and #4). Lastly, we simulated 800 targets (comparable to the sgRNA-3 design, 837 targetable sites) and applied the site editing bias. We defined accuracy as the percentage of correct pairwise relationships between the edits generated within G generations."

Also, we added the sentences below to the revised manuscript (page 9, lines 195–209).

"We conducted simulation experiments to check the validity of our method for long-term lineage tracing (**Figure 3**, see Methods). First, we performed simulation experiments assuming that dropouts occurred as the editing continued until a given generation (x -axis). The reconstruction accuracy increased as the number of traceable generations increased for a given number of editing sites. It reaches a peak when the editable sites were almost used up. At this turning point, >60% of the editable sites were consumed for all analyzed cells, resulting in ambiguous conclusions about phylogenetic relationships after this peak point and diminishing returns. Notably, if the editing rate increased, this peak shifted to an earlier generation. In contrast, slower editing rate was not saturated until the last generation. Also, increasing the number of targetable loci improved the reconstruction accuracy. After applying a 30% dropout rate across the target, the accuracy dropped by an average of 6.5% ($F=0.05$, close to our experimentally determined value of 0.06), 6.7% ($F=0.1$), 6.6% ($F=0.2$), and 5.7% ($F=0.5$), respectively, for each editing rate. When we further simulated the effect of site editing bias (**Supplementary Figure 12**), we observed no difference or a decreased difference (<1%) in accuracy."

Figure 3. Performance simulations of the targeted deaminase system. Continued editing was allowed until G generations. After that, the accuracy (y-axis) was calculated as the percentage of correct order of pairwise relationships between the edits generated within G generations. We compared the effect of dropouts on accuracy. (a) No dropout shown by solid lines and (b) dropout applied shown by dashed lines. The inflection point is indicated by the black arrow.

Supplementary Figure 12. Simulation of site editing bias (N=800). The site editing bias is simulated with 800 targets for comparison with an sgRNA-3 design (837 sites). We applied an empirically determined upper bound of the editing bias = 10% (~10% of the sites tend to saturate faster).

We also added following summary sentence to the Discussion section on page 10, lines 225–228.

“The simulation experiments provided the rationale for long-term lineage tracing and showed a robust performance even when site editing bias was present at the targets. When considering dropout effects in a simulation study, these findings illustrate that lineage tracing with our base editing system may have advantage over conventional Cas9-based system.”

3) How many L1 elements can the authors detect with their strategy, how many of those can be uniquely identified using the read sequence, and what fraction is typically detected in a single cell? It would be important to explicitly state these numbers to allow the reader to assess the advantages and limitations of the system. Please comment on the probability of the same site being edited independently in different cells.

There were 126,810 targetable L1 elements in our design (PCR with a pair of primers). Among these, 17,956 sgRNA spacers had at least two perfect matched sites in targetable L1 elements. The design for the single-cell lineage tracing is rank3 (ranked by the number of perfect target sites, which was 837 sites), designated as sgRNA-3. On average, 92.1% of the targetable sites [0.3 standard deviation (stdev)] were covered for all single cells amplified by PCR. Of those uniquely aligned reads and out of 837 possible edit sites for sgRNA-3, we detected 6.4 sites (3.3 stdev) (0.8%) in a single cell tree (Figure 2d). For the other two trees in Supplementary Figure 9, we detected 20.3 sites (9.3 stdev) (2.4%) and 9.0 sites (4.9 stdev) (1.1%). We summarized this result and added it to page 8, lines 174–176.

“On average, 92.1% [0.3 standard deviation (stdev)] was covered for all single cells amplified by PCR. Due to the stochastic nature of the edits, the number of edited sites varied from 6.4 to 20 (0.8–2.4%, stdev; range 3.3–9.3).”

In order to calculate the probability of the same site being edited independently in two different cells (Cell_1, Cell_2), assuming a specific shared edit site s , we can state this as a following equation:

$$P_I(s) \propto P_s^2 \times \prod_{i \neq s} P_i \times \prod_{j \neq s} P_j$$

Where $P_I(s)$ is the probability of two clones sharing the edit site s that was created independently in two different cells having i and j edit sites, respectively, and where P_s is the site specific (s) edit probability.

A site-specific edit probability (P_s) can be obtained from calculating the average of the per site editing efficiency for all analyzed cells (see below table for example).

	Site_1	Site_2	Site_3
Cell_1	0.5	0.1	0.3
Cell_2	0.2	0.05	0.35
Cell_3	0.1	0.2	0.1
Average	0.27	0.12	0.25
Normalized	0.42	0.18	0.40

For illustrative purposes, we calculated the average per site editing efficiency for three cells (0.27, 0.12, and 0.25) and normalized these values to 1 (by making the sum of the per site probabilities equal to 1) to obtain site specific probability (0.42, 0.18, and 0.4, shown in the above table). The normalization process estimates the contribution of each editing site across all cells to decide the cutoff value for determining which sites are used in lineage reconstruction. Higher value of normalized probability implies greater chance of having shared the same edits (from above equation).

Although differential editing among different sites is a potential confounding factor in the lineage reconstruction analysis, the overall negative effect on the reconstruction accuracy was minimal

(less than ~1% for all generations, see simulation result in comment #2 when we apply the site editing bias parameter).

4) Are there differences in editing efficiency between the different sites? This would be important to estimate the probability of the same site being edited in two different cells, which would complicate lineage tree reconstruction.

In line with the comment #3, the editing efficiency was different across the interspersed repetitive target regions. We have modified and added details about these methods in the following sentences on page 17, lines 396–406.

“First, we tabulated the distribution of the editing efficiencies across the target regions. Then, we normalized the per edit site average editing efficiency to a value of 1 by aggregating all sites and calculating the contributing fractions of each edited sites (see comment #3). These site edit probabilities (per site) were strongly correlated ($R^2 > 0.7$) between two cell lines. Since single-cells have different read structure compared to bulk cells, we designate the probability of bulk cells to single cells for the shared edit sites only.

To reduce the false reconstruction of the lineage, we excluded the sites with an average editing efficiency greater than 80% (normalized site editing probability > 0.004 , ~10% of the target sites were removed) because they were mostly used up, in other words, shared by almost every clone, in lineage reconstruction. Also, we removed highly correlating features (or edited sites) ($R^2 > 0.8$) to reduce the chance of misplacing the nodes where two cells might accidentally share edits.”

5) The authors should explain in the main text why they choose to use the iterative graph approach for the bulk experiments and hierarchical clustering for the single cell experiments.

An iterative graph approach is useful for analyzing complex network relationships. For example, a trifurcation network can be represented as a combination of bifurcation operations. For the controlled bulk experiments, the nodes for this controlled experiment were designed to contain more than two daughter nodes. Moreover, we observed iterative graph approach to be more robust for dropout events when certain cells fail to be observed. In contrast, time-lapse imaging follows a

typical ideal bifurcation lineage, and so we used a hierarchical binary clustering algorithm for a more robust result.

6) The authors should indicate in figure 2b which three nodes are misplaced and explain why they think this happened.

We appreciate the reviewer's concern. To clarify, we placed arrows above the misplaced nodes in Figure 2b. All of the misplaced nodes are placed in the last generation of the tree.

We added the following explanation on page 7, lines 146–157 and added colors corresponding to each clade in the legend of figure 2.

“For Clade1, the edits from the third generation of the tree accumulated different edits that matched Clade2. However, for Clade4, the misplaced nodes accumulated edits defined by the early edits in Clade3 and caused barcode mixing (sharing barcodes) between the clades. For the HeLa cells, we also observed slightly improved performance in accuracy (88% and 93% for conventional variant calling vs. custom algorithm, respectively) (**Supplementary Figure 7**). The reason for the clade mixing was the same as above for HEK293T cells. Although the probability of sharing independent edits from different lineages is low (please see response #3), 2-3 weeks between generations was sufficient time to acquire editing events from a neighboring lineage because lower allele frequencies limited sensitive variant calling for earlier lineages. These edits may have accumulated during this time and appeared in later lineages, thereby confounding lineage reconstruction. In addition, errors can also be introduced from sequencing at the transition between generations 3 and 4.”

Minor comments

7) The authors should explain which characteristics of L1 elements make this a good lineage tracing system, such as number of elements and sequence similarity between different elements. It's also unclear to me why the amplicon size was expected to follow a bimodal distribution (Results, first paragraph).

We agree with the reviewer's comment that the term bimodal distribution might confuse the reader because of the phrase that starts with "As expected". To clarify, this was an observed phenomenon that we do not expect. Depending on the primer design, we expected the characteristics of the amplified region to be different. We apologize for this and deleted the expression "As expected".

The rationale for choosing the L1 element is stated in the last paragraph of the Results section, under “Characterization and selection of target regions for the cell barcoding system (page 3)”. To add more detailed information about the target, we have added the following sentences on page 4, lines 79–83.

“There were 126,810 targetable L1 elements in our design (PCR with a pair of primers). Of these, 17,956 spacers had at least two perfect matched sites in the targetable L1 elements. In addition, when we investigated the top 100 targetable L1 elements, the average pairwise similarity was 28% and each had a high number of perfectly matched sites (range: 95-1585).”

Below is the figure related to this response. Similarity is scaled to 1 with the diagonal indicating internal similarity. We plotted only the top 100 targetable L1 elements because of the time complexity of the calculation required for all pairwise comparisons.

8) The figures should be improved to increase precision: Can you indicate cell sorting and approx. number of cell divisions in Fig. 2b? Why do the barcodes have different widths in Fig. 1a (orange and yellow bars)? And can you please highlight in Fig. 1a which sites are editable?

We appreciate this suggestion and have included additional figure legends in Fig. 2b. Please also see response #12 for additional figures. To clarify, we re-drew the Fig. 1a to have same width for each edit and also matched the editable sites with the barcode combinations (each barcode combination should be unique to each cell). The revised part is boxed in dashed red lines below.

Fig. 1a
Before

After

We have modified the widths of the confusing barcodes to the same extent. The editable part of the genome in each cell is marked in yellow and the corresponding part (marked in yellow) is composed of the barcode parts of each cell.

Fig. 2b
Before

After

To indicate the cell sorting and approximate number of cell divisions, the sorting method and the period of sorting are displayed. To increase visibility, the color of each clade is displayed, and each clade is numbered. In addition, the three misplaced nodes are indicated by red arrows.

9) The authors show in Fig. 1d that the editing efficiency of the two editable Cs of sgRNA3 is highly correlated, but they don't discuss the consequences or interpret this observation. Do you consider the two Cs in a target sites as separate entities for lineage tracing, or not?

We thank the reviewer for highlighting this point. Generally, base editing is processive when multiple Cs are present within the editing window (PMID: 27096365), which was consistent with our study. Thus, we treated two Cs in the target sites as one entity for lineage tracing. Editing efficiency was calculated as the average of the editing efficiency of the two Cs's. To clarify this issue, we added the following sentence to page 5, lines 103–105.

“which means base editing that converts most Cs in the editing window is processive. Thus, we treated these two Cs as a single editing site, by averaging the editing efficiency for the lineage tracing”

10) How was the amplicon size measured?

Amplicon size was measured as a fragment length, with major peaks at 124 and 142 bp. We have revised the legend for Supplementary Figure 2 accordingly.

11) Main text, line 87: Are these just mapped reads? How did authors control for PCR biases?

Yes, they are uniquely mapped reads that are properly paired. To control for PCR biases, we adjusted the input amount and the number of PCR cycles. We also integrated the molecular barcode sequence to precisely quantify each molecule. We have added the following paragraphs to page 3, lines 64–71.

“To compare the effect of precise molecular tag counting, we compared the effect of introducing degenerate bases (barcode) at the 5' end of one of the primers. We uniquely aligned an average of 30% of tags (with at least 50% of the terminal base sequences of the amplification primer being correct) to a unique genomic position and obtained 25,933 and 26,347 distinct aligned positions (dynamic fold range of 1.5 and 1.3) for non-barcode and barcoded samples, respectively (four replicates each). The uniformity (coefficient of variation) was higher in the barcoded sample (0.58 compared to the 0.82 for non-barcoded), minimizing the possibility of over-counting the true number of molecules in the final tally for each target bed region.”

12) Main text, line 114: The experimental design is poorly explained, and it is not clear what process is repeated. How exactly does the sorting work? What is its purpose? Is it about monoclonal expansion of positive cells in order to reduce mosaic effects? If so, authors should clarify and justify the experimental design. Directing only to a tree (2b) is not informative.

We thank the reviewer for raising important points and we agree that an additional figure is needed to clarify the *in vitro* tree expansion experiment, which we have included as Supplementary Figure 6. After isolating a positive cell, the expansion of these cells for 2–3 weeks results in a mosaic effect, rather than creating a homogeneous population. This allows reconstruction of the lineage with minimal loss of accuracy.

13) Main text, line 181: The comparison of editing efficiency compared to Cas9 is not convincing. The authors compare to Cas9 injection into zebrafish embryos, which is a very different system, so it's likely that the difference in dynamics is not due to their system per-se but to the delivery of the recorder.

We completely agree with the reviewer's perceptive comment, which is a salient point that needs to be discussed in detail. The previous barcoding systems have used Cas9 ribonucleoprotein (RNP) and *in-vitro* transcribed sgRNA that can be delivered to the cells by a direct injection method. However, in our barcoding system, we used plasmids containing engineered Cas9 (BE3) and sgRNA and used transient transfection such as lipid-based transfection or electroporation to introduce the vectors into the cells. As described in previous studies (PMID: 27229144, 29644996), RNP can be an efficient way to increase editing specificity *in vivo* with rapid clearance. Meanwhile, the local concentrations of Cas9 protein and sgRNA are important factors that affect the efficiency of genome editing events. For long-term experiments, expression of Cas9 using a plasmid-based system would be a good solution for long-term editing. We have rewritten the discussion to address this point, on page 10, lines 220–225 as below:

"From the perspective of long-term lineage tracing (0.06 edit per hour for our method vs 0.4 edit per hour for the conventional Cas9-induced method), our approach enables the editing of targets slowly but continuously, because PiggyBac-based plasmid delivery lasts longer inside the cells. However further studies that provide a direct comparison to previously established barcoding systems using sgRNA-Cas9 ribonucleoprotein are needed."

Reviewer #2 (Remarks to the Author):

The authors Hwang et al present here a straightforward application of the Cas9 deaminase system for use as a lineage tracing tool. The authors show in a stepwise fashion the progress from proof of concept in which they test by transient transfection two different gRNAs targeted to the L1 repeat element in the human genome, to application of this tool by PiggyBac transposons in bulk populations, and finally to single cells lineages. They use unique barcodes and a custom algorithm to build accurate trees, which they validate with “ground truth” time-lapse image data.

The authors' work is comprehensive, logically consistent, clearly described, and represents a step forward in the development of lineage-tracing tools. Comments below are minor in nature and could help the authors strengthen their work further.

We appreciate the reviewer's succinct and accurate summary of our manuscript. Thank you for the kind statements. We addressed all the comments and revised our manuscript.

1) How many L1 sites are available in the genome? Given the measured mutation rate, how many generations do the authors expect to be able to trace with this target set? It may also be useful to use their model to more generally show the number of generations traceable as a function of mutation rate and size of the target pool. Is there any risk, as the target space begins to saturate, that barcodes from different lineages will begin to collapse into the same, all-Cs-to-Ts in the same position at each target site, shared barcode?

We agree with the reviewer's comment regarding the further simulation of the lineage tree with multiple parameters. Please refer to comment #2 of reviewer #1 for traceable generation with editing rate and target size as a parameter. Also, for the risk of shared barcode issue, please see response #3 and #4 to reviewer #1 (the sites that are saturating at a faster rate were removed to reduce this risk). We also showed on simulation (comment #2 of reviewer #1) with site editing bias parameter that the accuracy drop was minimal (<1%).

2) Supplementary Figure 5 – please clarify by showing with a target sequence which Cs were or were not mutated.

We indicated the target Cs (4–8 bp window) and non-target (non-mutated) Cs for clarity, as per the reviewer's suggestion. We added the figure below as Supplementary Figure 5 in the revised supplementary manuscript (Mutated C's are underlined and Non-mutated C's are indicated inside the plotting box). We also added the sgRNA-1 sequence including each C position in the Supplementary Figure 5 legend as following sequences: [sgRNA-1 : 5'-ATGGGTG **C(1)** AG **C(2)** AAA **C(3)** **C(4)** A **C(5)** **C(6)** A-3'].

3) The authors describe their system as dCas9-based in their abstract and introduction, but their vector map (Supplementary Figure 4) and the BE3 system originally described by reference #18 make use of nickase Cas9.

Thank you for bringing this to our attention. We revised the dCas9 to nCas9 in both the abstract and introduction sections, as per the reviewer's suggestion (highlighted in the manuscript).

Reviewers' Comments:

Reviewer #1:

Remarks to the Author:

The authors have improved their manuscript, and I particularly appreciate the inclusion of simulations in Fig. 3. Below please find a few suggestions that may help improve the manuscript further.

I'm a bit concerned about the authors' reply to my comment #6. I had asked why some nodes are misplaced in Fig. 2b. The authors answered that the corresponding cells had acquired barcodes from different clades. The authors argued in their reply to my comments 2-4 that it is very unlikely that their results are affected by the same edit happening twice, in two independent events, and they attribute these errors to sensitivity issues in variant calling at earlier stages. I do not fully understand this argument, and I would request a more detailed diagnosis of why these cells are placed incorrectly in the tree. It's hard to see how this method may potentially be used in a mouse if a considerable number of cells are placed incorrectly, even in a very controlled cell culture system with low complexity and only three generations. As a part of this analysis, the authors should show which specific sites are edited in the individual nodes. Furthermore, they should substantiate their claim that edits that were initially below detection threshold accumulated during the 2-3 week expansion time – if this is true, some sub-threshold accumulation of the corresponding edits should already be detectable at earlier stages.

The Discussion needs to be improved by including a more detailed discussion of the limitations of the presented analysis. The authors argue in their reply to my comment #1 that for a proof of concept analysis it is not necessary to record deeper trees. In principle I agree with this statement, but it would be important to discuss more clearly that the approach has only been tested for 3-4 generations, and that even for such simple lineage trees errors in lineage reconstruction were observed.

The new Figure 3 should be explained better. What are the units of the cell division rate? Editing rate per hour? Per site and cell division? Also, it should be stated explicitly how the lineage trees were reconstructed from the simulated data, and the authors should illustrate how the accuracy is calculated. Furthermore, the legend should state that the colors correspond to different numbers of target sites. It makes sense that tree reconstruction fails when approaching saturation, but I don't understand why more generations (i.e. more complex trees) lead to better reconstruction during the early generations (left part of the graphs) – this should be explained in the manuscript.

I'm a bit confused about the authors' reply to my question #11. If I understand correctly, they introduced random barcodes (as unique molecular identifiers) in their PCR primers. How do you make sure that the random barcodes don't get overwritten at each PCR cycle? Also, the explanation in lines 65-71 is unclear: What is "dynamic fold range"? And what is a "target bed region"?

Please include the clarification of the experimental design (reply to my comment #12) as a figure panel.

Line 42: Toxic effects can also come from having APOBEC on for extended periods of time, which might affect lineages as well.

Reviewers' comments:

Reviewer #1 (Remarks to the Author):

The authors have improved their manuscript, and I particularly appreciate the inclusion of simulations in Fig. 3. Below please find a few suggestions that may help improve the manuscript further.

→ We appreciate the reviewer's additional suggestions for improving our manuscript. We addressed the concerns that were raised and trust this has strengthened the revised manuscript.

1. I'm a bit concerned about the authors' reply to my comment #6. I had asked why some nodes are misplaced in Fig. 2b. The authors answered that the corresponding cells had acquired barcodes from different clades. The authors argued in their reply to my comments 2-4 that it is very unlikely that their results are affected by the same edit happening twice, in two independent events, and they attribute these errors to sensitivity issues in variant calling at earlier stages. I do not fully understand this argument, and I would request a more detailed diagnosis of why these cells are placed incorrectly in the tree. As a part of this analysis, the authors should show which specific sites are edited in the individual nodes.

→ We apologize that the reviewer found our explanation unclear. As a result of the analysis, we added a figure to depict the edited barcodes in the target region. For most nodes, different clades could be clearly detected, although there were some overlapping barcodes (circled below the figure [Barcodes from *in vitro* bulk HEK293T cell experiments]). Thus, according to the reviewer's suggestion, we thoroughly investigated the issue to determine whether the discrepancy could be the result of a problem in the algorithm (finding the connected components in the depth-first search-based algorithm to initially assign clade identity).

Previous Figure 2b

We depicted the edited barcodes in the figure below for the nodes indicated by red arrows in previous Figure 2b.

Barcodes from *in vitro* bulk HEK293T cell experiments

Barcode representation for HEK293T cells indicated in **Figure 2b**. Rows correspond to the nodes for each clade in **Figure 2b** (node numbers shown on the right). The 'depth' of the tree for each clade is indicated by color gradients (lighter shades correspond to deeper tree nodes in **Figure 2b**). Columns correspond to editing sites in the sgRNA-3 design. Shared barcodes associated with misplacement of the nodes in the clades using the previous depth-first search-based algorithm are indicated by dotted-line circles.

Previous Supplementary Figure 7

We depicted the edited barcodes below for the nodes indicated by red arrows in previous **Supplementary Figure 7**.

Barcodes from *in vitro* bulk HeLa cell experiments

In the case of HeLa cells shown above, the dotted-line circles indicate the shared barcodes associated with misplacement of the nodes in the clades using the previous depth-first search-based algorithm.

Following the reviewer's suggestion, we modified the algorithm, which resulted in improvement of the reconstruction accuracy. To explain these changes in the method, we revised the Methods section (see page 19, lines 431–449).

- “The graph reconstruction strategy first employs a depth-first search (DFS) to identify the strongest connected components using edited sites. As opposed to the conventional algorithm used to identify connected components, we initially employed a DFS approach to maximize the weight of connected components (the sum of the sequencing depths of the component is maximized). As the graph search gives priority to high-depth components, the DFS-based algorithm resulted in a few nodes inadvertently being placed on different clades, as the depth of shared edits was unusually high. Therefore, we modified the algorithm to identify connected components based on the overlapping fraction of edited barcodes between nodes (nodes in the same clade share a higher fraction of barcodes than with nodes in other clades). In so doing, no errors were introduced in distinguishing the clades. Only minimal error (accuracy increased to 97% for experiments involving both HEK293T and HeLa cells) occurred in assigning the mother-daughter relationship within clades. Assigning the correct in-clade mother-daughter relationship depends largely on the continuous accumulation of edited barcodes. Thus, the remaining error appears to be due in part to the nature of the bulk sequencing results, in that some PCR or sequencing error contributes to the final barcode combination of specific nodes, thus leading to misassignment of the subtle mother-daughter relationship. For example, if a node (daughter) from one mother node got the edits that should belong to other node (daughter) from another mother node by chance

resulting in more overlap with another mother node, the algorithm will likely place the node in the wrong place because assignment of the mother-daughter relationship depends on how much the barcode combination is shared with the ancestor.”

We also added detailed explanations regarding the results below; see page 7, lines 152–155.

“We note that the remaining errors might confound lineage reconstruction due in part to the nature of the bulk sequencing experiment, because some PCR or sequencing errors could have contributed to the final barcode combination of a given specific node (see **Supplementary Figure 8** and **Methods**).”

The figures shown below (**Figure 2b**, **Supplementary Figure 7**, and **Supplementary Figure 8** [with legends]) are updated versions incorporating the improved algorithm.

New Figure 2b

Furthermore, we included the following statement in the legend for **Figure 2b**: “The red solid line connects the incorrectly placed mother-daughter node, and the dotted line indicates the correct mother-daughter node connection.”

New Supplementary Figure 7

Furthermore, we included the following statement in the legend for **Supplementary Figure 7**:
 "The red solid line connects the incorrectly placed mother-daughter node, and the dotted line indicates the correct mother-daughter node connection."

“Supplementary Figure 8. Barcode representation of *in vitro* bulk cell experiments (from top, HEK293T and HeLa cells). Rows correspond to nodes for each clade. The clade to which each row (node) belongs is shown to the left; node numbers shown in Figure 2 and Supplementary Figure 7 are shown to the right for each row. The “depth” of the tree is indicated by a color gradient (lighter shades correspond to deeper tree nodes in Figure 2b and Supplementary Figure 7, and node numbers shown in red font indicate misplaced nodes). Columns correspond to editing sites in the sgRNA-3 design. Boxes shown in black dotted lines indicate cell barcodes from misplaced nodes.”

The updated accuracy information is highlighted on page 7, lines 148–152.

“Compared to the conventional variant calling approach, we observed an average of 70 cell barcodes for sgRNA-3 and the reconstruction accuracy improved to 97% (35/36) (Figure 2b). For the HeLa cells, we also observed slightly improved performance in accuracy (88% and 97% (59/61) for conventional graph searching and variant calling vs. the custom algorithm, respectively) (Supplementary Figure 7).”

Furthermore, they should substantiate their claim that edits that were initially below detection threshold accumulated during the 2-3 week expansion time – if this is true, some sub-threshold accumulation of the corresponding edits should already be detectable at earlier stages.

→ I hope I correctly understand the reviewer’s comment. With regard to shared edits that caused misplacement of the nodes, the average depth of coverages was consistently below 3 (<3x, allele frequency of 0.3% [variant depth/total depth of coverage]), making it unlikely to lead to identification as a variant in the earlier stages (even though these supporting reads are visible in IGV). Accordingly, we removed our initial claim of “accumulating edits”.

It’s hard to see how this method may potentially be used in a mouse if a considerable number of cells are placed incorrectly, even in a very controlled cell culture system with low complexity and only three generations.

→ We thank the reviewer for pointing out the issue of misplaced nodes. In response to the reviewer’s suggestion, using the improved algorithm, we increased the reconstruction accuracy to 97%, an increase of 5% compared with the previous reconstruction accuracy (92%). Although we did observe a few errors (~3%) in the controlled tree experiments, we anticipate being able to reduce the rate of errors by simultaneously using different sgRNAs. We revised the Discussion section to address this issue (also, please see response to point #2 below). As described in a recent study (PMID:30093604), multiple sgRNAs can be introduced simultaneously into synthetic interspersed target sites to increase the complexity of the barcodes, which could facilitate lineage reconstruction for multicellular higher organisms such as mice.

2. The Discussion needs to be improved by including a more detailed discussion of the limitations of the presented analysis. The authors argue in their reply to my comment #1 that for a proof of concept analysis it is not necessary to record deeper trees. In principle I agree with this statement, but it would be important to discuss more clearly that the approach has only been tested for 3-4 generations, and that even for such simple lineage trees errors in lineage reconstruction were observed.

→ As per the reviewer’s request, we revised the Discussion section as shown below to discuss the limitations of our study more clearly (see page 11, lines 248–253).

“We acknowledge that our current platform has been tested over a limited number of generations (up to four generations). In the *in vitro* tree expansion experiment, we observed some errors (~3%) in the reconstruction of the tree. However, we anticipate that this negative effect on reconstruction accuracy would be attenuated by simultaneously introducing multiple sgRNAs into the interspersed target sites, as this would increase the complexity of the barcodes and potentially facilitate lineage reconstruction for multicellular higher organisms such as mice.”

3. The new Figure 3 should be explained better. What are the units of the cell division rate? Editing rate per hour? Per site and cell division? Also, it should be stated explicitly how the lineage trees were

reconstructed from the simulated data, and the authors should illustrate how the accuracy is calculated. Furthermore, the legend should state that the colors correspond to different numbers of target sites. It makes sense that tree reconstruction fails when approaching saturation, but I don't understand why more generations (i.e. more complex trees) lead to better reconstruction during the early generations (left part of the graphs) – this should be explained in the manuscript.

- We thank the reviewer for the detailed suggestions regarding the simulation and corresponding **Figure 3**.

Regarding **Figure 3**, “the units for the editing rate (base substitution rate) are mutations/site/cell division” and this information is now included on page 21, lines 490–491 and in the legend for **Figure 3**. As the reviewer suggested, we also included the following statement in the **Figure 3** legend: “Different colors correspond to different numbers of target sites.”

To describe the reconstruction method in greater detail, we revised the text to include additional explanation of the reconstruction method (see pages 19–20, lines 431–464).

“The first part of the graph reconstruction strategy utilizes a depth-first search (DFS) to identify the strongest connected components using edited sites. Unlike the conventional algorithm for identifying connected components, we initially used the DFS to maximize the weight of connected components (the sum of the sequencing depths of the component is maximized). As the graph search gives priority to high-depth components, the DFS-based algorithm resulted in a few nodes inadvertently being placed on different clades, as the depth of shared edits was unusually high. Therefore, we modified the algorithm to identify connected components based on the overlapping fraction of edited barcodes between nodes (nodes in the same clade share a higher fraction of barcodes than with nodes in other clades). In so doing, no errors were introduced in distinguishing the clades. Only minimal error (accuracy increased to 97% for experiments involving both HEK293T and HeLa cells) occurred in assigning the mother-daughter relationship within clades. Assigning the correct in-clade mother-daughter relationship depends largely on the continuous accumulation of edited barcodes. Thus, the remaining error appears to be due in part to the nature of the bulk sequencing results, in that some PCR or sequencing error contributes to the final barcode combination of specific nodes, thus leading to misassignment of the subtle mother-daughter relationship. For example, if a node (daughter) from one mother node got the edits that should belong to other node (daughter) from another mother node by chance resulting in more overlap with another mother node, the algorithm will likely place the node in the wrong place because assignment of the mother-daughter relationship depends on how much the barcode combination is shared with the ancestor.

Inherently, the first ancestor node (bulk cells) had a maximal degree of connections with the other nodes. We removed this node and identified the ancestor (top) node from the remaining connected components in an iterative fashion. To identify the top node for the remaining components, we must calculate the detection rate $p(x)$ first, assuming x is a top edit. This can be calculated as the ratio of the number of cells (nodes) that express edit x and an edit connected to x to the number of cells (nodes) that express edits connected to x [$p(x) = N_{x \cap c(x)} / N_{c(x)}$, where $c(x)$ is a set of edits that are connected to edit x]. Thus, the chance of observing edits between edits x and y in at least one cell is calculated as $p(x-y) = 1 - (1 - p(x))^{N_y}$, where N_y represents cells (nodes) expressing edit y . As such, the underlying distribution of edit x is a *poisson binomial* with a different success probability defined as $p(x-y)$. Finally, we calculated the probability of measuring at most the observed degree using the distribution (if you have 4 nodes around the specific nodes but the actual observed

degree is 3, then you calculate the cumulative distribution function for at most 3 connections). We used the “*poissonbinom*” R package to calculate the probability density. The node with the highest probability of this value is considered the top node (see Supplementary Figure 20a in ref 7 (PMID: 29644996) for an illustrative example).”

The accuracy for the simulation is calculated as follows, (this text was added on pages 22, lines 504–509, of the revised manuscript):

“The accuracy of the simulation experiment was calculated as the fraction of triplets (a root node is divided into two daughters when the cell divides [the sum of mother and daughters]) that are correctly placed compared to the ground-truth mother-daughter relationship. For example, if we have a 3-depth tree (1+2+4=7 cells) with a total of 3 triplets and found 1 triplet that was correct, the accuracy would be 33% (1/3).”

We also added further explanation of the results shown in **Figure 3** on page 9, lines 198–203.

“For the earlier generations (up to the 9th generation) before it reaches the peak, we observed a sharp increase in accuracy because the process of accumulating barcodes was sparse, which means that there was insufficient sharing between sister clades (two descendants that split from the same node) to allow accurate placement. Moreover, as the process of mutation accumulation is exponential, when the fraction of shared barcodes between sister clades exceeded ~85%, the increase in accuracy was minimal (<2%) before reaching the peak.”

4. I’m a bit confused about the authors’ reply to my question #11. If I understand correctly, they introduced random barcodes (as unique molecular identifiers) in their PCR primers. How do you make sure that the random barcodes don’t get overwritten at each PCR cycle? Also, the explanation in lines 65-71 is unclear: What is “dynamic fold range”? And what is a “target bed region”?

→ As the random barcodes were attached by PCR, they can be overwritten at each PCR cycle, as the reviewer indicated. However, as we performed only a small number of PCR cycles using a primer containing random barcodes (as unique molecular identifiers, UMIs), the likelihood of an increase in the total population of L1-region PCR fragments with random barcodes is greater than changing the ratios of each random barcode. Subsequently, the number of the second PCR cycle for attaching index adaptors for sequencing was also reduced to minimize potential bias. We concluded that this would reduce PCR amplification bias (similar to the principle of the previous Safe-seqS method [PMID:21586637]). Therefore, our best option for reducing bias was to perform a limited number of PCR cycles. We added the following explanation on page 3, lines 62–64.

“We reduced PCR amplification bias by using two-step PCR with a primer containing degenerate bases according to a principle similar to that of a previous method (Ref-PMID:21586637) (see **Methods**).”

Also, as per the reviewer’s suggestion, we added a more detailed description of counting unique molecules to reduce bias (see page 17, lines 391–396).

“For precise molecule counting, sequencing reads sharing the same UMI (degenerate bases) were grouped into families and merged if $\geq 70\%$ contained the same sequence. In addition, to minimize the effect of over-counting the same molecules, we calculated the distances between UMIs; Hamming distances ≤ 2 were merged in the Hamming-distance graphs. We only retained UMIs exhibiting the highest counts within the clusters.”

We also revised the ambiguous wording on pages 3–4, lines 66–71.

“To compare the effect of precise molecular tag counting, we compared the effect of introducing degenerate bases (unique molecular identifier, UMI) at the 5' end of one of the primers. An average of 30% of tags (with at least 50% of the terminal base sequences of the amplification primer being correct) were uniquely aligned to a reference genome, and we obtained 25,933 and 26,347 distinct aligned positions (average number of uniquely aligned positions from four replicate experiments for non-barcode and barcoded samples, respectively).”

We removed the term “dynamic fold range,” as this was confusing.

To avoid confusion, we also removed the word “bed” from “target *bed* region” (line 74).

5. Please include the clarification of the experimental design (reply to my comment #12) as a figure panel.

→ As per reviewer’s suggestion, we revised the Figure to clarify our experimental protocol. Thus, we added a detailed experimental design to the panel in **Supplementary Figure 6** (reply to your previous comment #12) as follows:

“Supplementary Figure 6. Schematic overview of *in vitro* tree expansion in the bulk experiment. First, the barcoding system containing BE3 and sgRNA is transduced into cells using lentivirus and selected using puromycin. Single cells are isolated from mCherry-expressing cells in individual wells using FACS. Isolated positive cells are expanded for 2–3 weeks, leveraging a mosaic effect rather than making a homogeneous population. This facilitates reconstruction of the lineage with minimal loss of accuracy. Some of the expanded cells are used for gDNA extraction and barcode amplification. Each cell barcode is then detected at the population level using NGS. The rest of the expanded cells are subjected for isolation of single cells using FACS, and these are expanded as in the previous procedure. The process is repeated over several generations, and cell barcodes are continually detected during this process. Reconstruction of the cell lineage is performed using the obtained cell barcodes.”

1. Barcoding system transfection and puromycin selection
2. mCherry-positive single cell isolation by FACS
3. The cell expansion for 2~3 weeks in individual wells
- 4-a. Some of the expanded cells' barcode detection at population level through gDNA extraction and barcode amplification
- 4-b. The remainder of the expanded cells is repeated over several generations of the 2~4-a process.
- ⋮
- N. The accurate reconstruction of cell lineage using the resulting cell barcodes of each generation

6. Line 42: Toxic effects can also come from having APOBEC on for extended periods of time, which might affect lineages as well.

→ We thank the reviewer for pointing out this issue; we revised the Introduction section accordingly (see page 2, lines 40–42).

“We hypothesized that CRISPR/Cas9-induced double-strand breaks (DSBs) in these numerous endogenous interspersed sites present in the genome would result in the loss of useful information.”

We also acknowledged APOBEC would be toxic to cells when expressed for extended periods due to the introduction of unwanted substitutions. We added the following sentence in the Discussion section (see page 11, lines 253–255):

“Furthermore, targeted deaminase can be cytotoxic if continuously expressed within cells, and this requires additional investigation.”

Reviewers' Comments:

Reviewer #1:

Remarks to the Author:

I thank the authors for incorporating my additional comments. The manuscript has improved in clarity, and I would recommend publication of this proof-of-concept study without further changes.

REVIEWERS' COMMENTS:

Reviewer #1 (Remarks to the Author):

I thank the authors for incorporating my additional comments. The manuscript has improved in clarity, and I would recommend publication of this proof-of-concept study without further changes.

** See Nature Research's author and referees' website at www.nature.com/authors for information about policies, services and author benefits

This email has been sent through the Springer Nature Tracking System NY-610A-NPG&MTS